# The Role of Non-Selective TNF Inhibitors in Demyelinating Events

**DOI:** 10.3390/brainsci11010038

**Published:** 2021-01-01

**Authors:** Line Buch Kristensen, Kate Lykke Lambertsen, Nina Nguyen, Keld-Erik Byg, Helle H Nielsen

**Affiliations:** 1Department of Neurology, Odense University Hospital, J.B. Winsloewsvej 4, 5000 Odense C, Denmark; line.marie.buch.kristensen@rsyd.dk (L.B.K.); klambertsen@health.sdu.dk (K.L.L.); keld-erik.byg@rsyd.dk (K.-E.B.); 2Department of Neurobiology Research, Institute of Molecular Medicine, University of Southern Denmark, J.B. Winsloewsvej 21, St., 5000 Odense C, Denmark; 3BRIDGE—Brain Research—Inter Disciplinary Guided Excellence, Department of Clinical Research, J.B. Winsloewsvej 19, 5000 Odense C, Denmark; 4Department of Radiology, Odense University Hospital, J.B. Winsloewsvej 4, 5000 Odense C, Denmark; nina.nguyen@rsyd.dk; 5Rheumatology Research Unit, Odense University Hospital and University of Southern Denmark, Odense University Hospital, J.B. Winsloewsvej 4, 5000 Odense C, Denmark

**Keywords:** TNF inhibitor, infliximab, demyelination

## Abstract

The use of non-selective tumor necrosis factor (TNF) inhibitors is well known in the treatment of inflammatory diseases such as rheumatoid arthritis, Crohn’s disease, and psoriasis. Its use in neurological disorders is limited however, due to rare adverse events of demyelination, even in patients without preceding demyelinating disease. We review here the molecular and cellular aspects of this neuroinflammatory process in light of a case of severe monophasic demyelination caused by treatment with infliximab. Focusing on the role of TNF, we review the links between CNS inflammation, demyelination, and neurodegenerative changes leading to permanent neurological deficits in a young woman, and we discuss the growing evidence for selective soluble TNF inhibitors as a new treatment approach in inflammatory and neurological diseases.

## 1. Introduction

The use of tumor necrosis factor (TNF) blockers has revolutionized the treatment of a number of chronic inflammatory diseases such as Crohn’s disease, ankylosing spondylitis, psoriasis, and rheumatoid arthritis.

Although TNF blockers are generally considered safe, an increasing number of neurologic adverse effects have been reported in the literature, consisting primarily of demyelination of the central nervous system (CNS) or peripheral nervous system (PNS) with a prevalence ranging from 0.050 to 0.100% [1]. These adverse events suggest a possible relationship between the use of anti-TNF biologics and demyelinating diseases [2].

TNF is a pleiotropic cytokine known to play important regulatory roles in the development and homeostasis of the healthy CNS [3]. It is produced initially as a transmembrane molecule (tmTNF) and is subsequently released from the cell as a soluble cytokine (solTNF) via regulated cleavage of tmTNF by TNF-α converting enzyme (TACE). Under normal conditions, TNF can be produced in the periphery by monocytes/macrophages, lymphocytes (T and B), natural killer cells, and dendritic cells [4], while TNF in the CNS is produced mainly by microglia [5]. Both forms of TNF are biologically active and interact with two distinct receptors—TNFR1 and TNFR2—with TNFR1 being expressed in all cell types, whereas TNFR2 is expressed mainly on immune cells, oligodendrocytes, and endothelial cells [6]. solTNF has a high affinity for TNFR1, which contains a death domain and can mediate apoptosis and chronic inflammation [7]. In the CNS, tmTNF has a higher affinity for TNFR2 and promotes mostly protective features such as cell survival and remyelination [6,7].

In animal models of multiple sclerosis (MS), such as experimental autoimmune encephalomyelitis (EAE), the administration of TNF blockers slowed the demyelinating process and improved outcome [8,9]. Ablation of TNF or TNFR1/TNFR2 combined in mice did not protect from EAE, however, and instead caused exacerbation of chronic disease [10,11,12,13]. TNFR1 ablation resulted in less severe EAE and better remyelination, while TNFR2 ablation exacerbated EAE and prevented remyelination [11,13,14]. Furthermore, mice only expressing tmTNF showed suppression of EAE [15]. These studies suggest a dichotomy between solTNF and tmTNF, in which MS is associated with the detrimental effects of solTNF via TNFR1, but tmTNF signaling via TNFR2 is important for repair and remyelination. This is supported by studies in EAE mice treated with a selective blocker of solTNF, XPro1595, which resulted in improved function, significant axonal preservation, oligodendrocyte differentiation, and remyelination [6,7]. More importantly, these differences in receptor function might explain the failed clinical trial using the non-selective TNF inhibitor Lenercept as treatment for MS [16], which was terminated due to clinical and radiological disease progression.

We present here a case of severe demyelination following treatment with infliximab, a chimeric monoclonal antibody that prevents binding of TNF to TNFR1 and TNFR2 by blocking both solTNF and tmTNF [17]. We discuss the possible underlying mechanisms of TNF blockers in CNS demyelination in the context of the current literature.

## 2. Case Description

The case is a 27-year-old woman with psoriatic arthritis, treated with methotrexate and infliximab. No family history of neurological disorders was reported.

After 4.5 years of treatment with infliximab, she complained of increasing fatigue, muscle pain, and mild cognitive difficulties. After a few months, she developed a subacute hemiparesis. MRI scan showed a solitary process with gadolinium ring enhancement (Figure 1—Week 0).

CSF analysis revealed no pleocytosis and normal protein levels, but did reveal oligoclonal bands and an elevated IgG index of 0.78. Initial blood tests showed a positive toxoplasmosis IgG, and PET-CT and a following biopsy from a single lymph node showed signs of granulomatous inflammation. The infliximab was discontinued, and the patient was initially treated for both toxoplasmosis and tuberculosis (Figure 1—Week 0).

Further investigations revealed no pathology on repeated PET-CT, including no pathological lymph nodes.

Repeated blood and CSF tests were found negative for toxoplasmosis, tuberculosis, varicella-zoster, Epstein–Barr, herpes simplex I+II, cytomegalovirus, Bartonella, Brucella, Aspergillus, hepatitis, and HIV. Microbiome PCR sequencing revealed no pathological DNA.

Antinuclear antibodies (ANA), Extractable Nuclear Antigen antibodies (ENA), Proteinase 3 Anti Neutrophil Cytoplasmic Antibody (C-ANCA), and Perinuclear Anti Neutrophil Cytoplasmic Antibodies (P-ANCA) were all negative.

Despite discontinuation of infliximab, the patient progressed clinically and radiologically over the next 12 weeks (Figure 1—Week 12). Clinical evaluation showed grade 4 hemiparesis on the left side and no cognitive difficulties, headache, or nausea.

Repeated CSF analysis confirmed oligoclonal bands and elevated IgG index but no increase in cell count or protein. Cytology described normal-appearing lymphocytes without pathological changes. Flow cytometry revealed 74% T cells, 7.9% polyclonal B cells, 1.6% NK cells, and CD4/CD8 of 4.1.

A biopsy was taken from the lesions in the brain, and histology showed unspecific reactive microglia and few infiltrating CD8+ T and B cells. There were no signs of tumor, lymphoma, vasculitis, progressive multifocal leukoencephalopathy, or infection with herpes, toxoplasmosis, or tuberculosis.

The patient was treated with high-dose glucocorticoid intravenously (1 g a day for 5 days) followed by five courses of plasma exchange (Figure 1—Week 12). Afterwards, MRI showed no new lesions but progression of one existing lesion and oedema, and the patient remained stable (Figure 1—Week 16). The patient continued with oral steroid therapy, which was tapered off after introduction of a two-course Rituximab treatment as a steroid-sparing agent (Figure 1—Week 16).

The following MRI showed remission, and the patient continued with a slight hemiparesis.

After 2 years (Figure 1—Week 104), the patient’s psoriatic arthritis started to deteriorate, and she was put on Secukinumab. She remained neurologically stable throughout the period and at follow-up (Figure 1—Week 134).

## 3. Discussion

We present a case of severe demyelination in a young woman who received infliximab as treatment for psoriatic arthritis. Because of the atypical presentation, many differential diagnoses were considered:

### 3.1. Infection

As the patient had been on different types of immunosuppressants since her early childhood, and at the time her symptoms developed she was treated with both infliximab and methotrexate, an infectious cause was first suspected. This was supported by the first MRI, which showed a characteristic ring enhancement comparable to cerebral toxoplasmosis, and a positive IgA test for toxoplasmosis. To further complicate the matter, a biopsy from a neck gland showed granulomatous inflammation, which could suggest tuberculosis. However, later test results for tuberculosis were negative (using interferon-gamma blood tests and microscopy of biopsy material). Although the patient received treatment for both toxoplasmosis and tuberculosis, this had no effect on the intracranial process or her clinical status. Furthermore, the CSF showed no signs of pleocytosis or elevated protein, nor did it confirm toxoplasmosis, tuberculosis, or any other form of infection.

### 3.2. Malignancy

Treatment with a TNF inhibitor has been suspected to increase the risk of certain malignancies, but this is still debated and may be related to the underlying autoimmune disease rather than the treatment [18,19,20,21]. On the initial PET-CT, the patient had a suspicious lymph gland. Biopsy from both this and a brain lesion showed no sign of malignant disease, just as repeated CSF cytology described normal-appearing lymphocytes without pathological changes. The clinical and radiological stabilization at 2 years follow-up also suggests a more benign disease process.

### 3.3. Inflammatory Demyelination

Studies suggest that patients with an autoimmune disease have an increased risk of developing other diseases with similar pathological mechanisms. Although still controversial, psoriasis has been shown to carry an increased risk of MS [22,23,24]. Reasons for this include shared genetic, environmental, and immunopathological factors (such as a strong Th1/Th17 response) in the two disorders [25]. This is further supported by the overlap in medications such as fumarate and Secukinumab [26]. As illustrated by the divergent results to anti-TNF inhibitors [16] however, the immunopathogenesis is not identical.

A register-based study from 2019 found an increased risk of MS and neuroinflammatory disease in patients with psoriatic arthritis but not rheumatoid arthritis, both with and without treatment with non-selective TNF inhibitors [27]. In view of the close relationship between autoimmune diseases, it is therefore possible that treatment with TNF blockers simply exacerbates a dormant neuroinflammatory condition like MS.

In this case, the patient remained stable without specific MS treatment, although Secukinumab was later introduced as an arthritis treatment and could in theory suppress any underlying MS activity. Secukinumab has been shown to be effective in MS [26], but it seems unlikely that an aggressive debut of MS would later stabilize completely without regular MS treatment. However, the disease course may still support a monophasic demyelinating event.

### 3.4. Infliximab-Induced Demyelination

As previously mentioned, the use of non-selective TNF blockers has been linked to demyelinating diseases, including MS, optic neuritis, transverse myelitis, and demyelinating neuropathies [1,28]. Furthermore, a potential link between TNF inhibitors and demyelinating disease has been suggested [29,30,31]. In view of the large number of patients treated with TNF blockers for long periods of time however, the demyelinating cases are rare and divergent [32,33]

The patient described here received infliximab for approximately 4.5 years before developing symptoms. In a 2017 review of reported cases of CNS demyelination associated with TNF inhibitors [26], the mean time of exposure was 17 months, but some patients had been treated for 6 years before developing symptoms. Our patient showed no resolution of her symptoms, consistent with 28% of the reviewed cases, and she had no prior family history of MS, which was also the case in the majority of cases reviewed. This patient thus shares many similarities with other cases of psoriatic arthritis treated with TNF blockers, supporting an initial monophasic demyelinating event that may later develop into MS, as seen in some cases with longer follow-up [28].

The mechanisms behind the potential demyelinating role of non-selective TNF blockers are not completely clear. In general, TNF blockers are large molecules that do not penetrate the intact blood–brain barrier, possibly explaining the lack of positive effects in TNF-mediated CNS diseases as opposed to other autoimmune disease such as rheumatoid and psoriatic arthritis [34,35]. Another theory is that the peripheral immunological effects of TNF blockers might alter the cytokine profile into a more proinflammatory CNS profile [28] or increase the ingress of peripheral autoreactive T-cells into the CNS through the leaky blood–brain barrier already established in active MS [34,35]

The infliximab used in this case blocks binding of TNF to TNFR1 and TNFR2 by blocking both solTNF and tmTNF [17], therefore also blocking the positive effects of TNFR2. Levels of TNF are increased in patients with MS, and studies suggest that while high levels of TNFR1 may be a risk factor for more severe disease, TNFR2 plays a more protective role [36]. As infliximab affects both pathways, the blocking of the protective features of TNFR2 and the remyelinating process could potentially lead to demyelinating events, as seen in our case. This is further supported by animal studies in which a selective blocker of solTNF resulted in improved function as well as significant axonal preservation, oligodendrocyte differentiation, and remyelination [6,7]. Therefore, selective TNF inhibition or activation of TNFR2 could lead to a new treatment approach for inflammatory disease [37,38].

## 4. Conclusions

This case supports the growing evidence suggesting demyelinating events as adverse effects of non-selective TNF inhibitors, but distinguishes itself by the long follow-up time and the in-depth discussion of the underlying mechanisms using selective versus non-selective TNF inhibition. As demyelination is a rare adverse event and TNF blockers are generally considered safe and effective in many chronic autoimmune disorders, this should not preclude this treatment, but rather suggests caution in patients of high risk of developing demyelinating events. With the introduction of selective solTNF inhibitors and the growing evidence of their effect, this may form the basis of a new treatment approach in inflammatory and neurological diseases.

## Figures and Tables

**Figure 1 brainsci-11-00038-f001:**
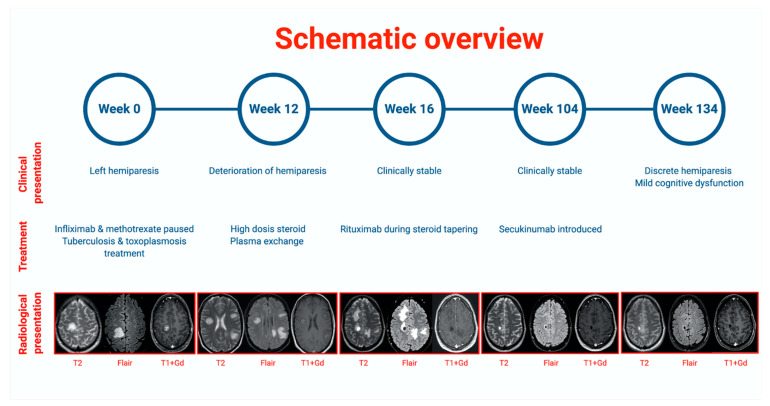
Schematic overview of the case, showing the relationship between clinical presentation, treatment, and radiological presentation shown by MRI scans (T2, Flair, and T1+gadolinium enhancement). Created with BioRender.com.

## Data Availability

The data presented in this study are available on request from the corresponding author. The data are not publicly available due to Danish data protection regulations.

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
