# Peer review of "The Role of Non-Selective TNF Inhibitors in Demyelinating Events"

_brainsci, 2021, doi:10.3390/brainsci11010038_

Round 1
Reviewer 1 Report
The case report "The role of non-selective TNF inhibitors in demyelinating events" by Kristensen et al is well written and interesting. I enjoyed reading your work but have two minor comments for you.
The role of non-selective TNF inhibitors in demyelination is known and there are case reports available in the literature. You have chosen to add your case report to this literature but I am missing a sentence or two that highlights what is unique with your case compare to what is already published. In which way is your work adding new information to the field.
In your last paragraph, third sentence there are two (that), you can delay one as the sentence needs only one.
Author Response
We would like to thank the reviewer for the constructive criticism and have revised the manuscript as follows:
We have included a conclusion which highlights the unique features of this case and how it adds new information to the field. The section now provides a take home message, page 7 section 4.
We have proofread the manuscript to remove redundant words and misspellings, particular with respect to page 6, last paragraph, line 3 (Reviewer #1)
Reviewer 2 Report
This is an interesting paper consisting of a brief review of the topic of demyelination and TNF alpha mabs.
The submission centres on a case report on a severe CNS demyelinating event accompanying the use of infliximab in a young woman with psoriatic arthritis. The authors use this to provide a brief review of the subject entitled: :The role of non-selective TNF inhibitors in demyelinating events.
The authors start off with a review of the function and role of TNF in general. They elicit the regulatory role of TNF with reference to the EAE animal model and its role in myelination .
The case report is clearly written providing relevant information pertaining to the case.
The MRI sequence is clear and demonstrates sequence of events nicely. In the discussion they then go on to discuss a wide differential diagnosis of the demyelinating event. This provides sufficient depth to the case report.
Overall the authors provide insight to the possible mechanisms behind the demyelinating event and TNF alpha blockers which are of use to the clinician and researcher. There are 2 minor issues that need addressing before this can be published.
Minor points:
The authors state that autoantibodies were looked for without providing details on which ones. It would be important to know which ones were looked for.
The paper ends without a conclusion or take home message.
I have one issue: The authors should explain what autoantibodies were specifically tested for instead of issuing a non explanatory statement: Autoantibodies were negative.
Author Response
We would like to thank the reviewer for the constructive criticism and have revised the manuscript as follows:
We have included a conclusion which highlights the unique features of this case as well as provides a take home message, page 7 section 4.
We have provided information on the autoantibodies that were investigated in the case, p 4, line 15-18.